# Is SARS-CoV-2 Directly Responsible for Cardiac Injury? Clinical Aspects and Postmortem Histopathologic and Immunohistochemical Analysis

**DOI:** 10.3390/microorganisms10071258

**Published:** 2022-06-21

**Authors:** George-Călin Oprinca, Lilioara-Alexandra Oprinca-Muja, Manuela Mihalache, Rares-Mircea Birlutiu, Victoria Birlutiu

**Affiliations:** 1Faculty of Medicine Sibiu, Lucian Blaga University of Sibiu, 550169 Sibiu, Romania; georgecalin.oprinca@ulbsibiu.ro (G.-C.O.); lilioaraalexandra.muja@ulbsibiu.ro (L.-A.O.-M.); manuela.mihalache@ulbsibiu.ro (M.M.); victoriabirlutiu@yahoo.com (V.B.); 2FOISOR Clinical Hospital of Orthopedics, Traumatology, and Osteoarticular, 030167 Bucharest, Romania

**Keywords:** SARS-CoV-2 infection, cardiac injury, clinical aspects, autopsy, histopathologic analysis, immunohistochemical analysis

## Abstract

Myocardial injury in patients with SARS-CoV-2 infection may be attributed to the presence of the virus at the cellular level, however, it may also be secondary to other diseases, playing an essential role in the evolution of the disease. We evaluated 16 patients who died because of SARS-CoV-2 infection and analyzed the group from both clinical and pathological points of view. All autopsies were conducted in the Sibiu County morgue, taking into consideration all the national protocols for COVID-19 patients. Of the 16 autopsies we performed, two were complete, including an extensive examination of the cranial cavity. In our study, the cardiac injury was primarily cumulative. Chronic cardiac injuries included fatty infiltration of the myocardium in five cases, fibrosis in 11 cases, and coronary atherosclerosis in two cases. Among the cases with evidence of acute cardiovascular injuries, inflammatory lymphocytic infiltrate was observed in nine cases, subepicardial or visceral pericardial neutrophil-rich vascular congestion in five cases, and venous thrombosis in three cases. Acute ischemia or myocytic distress was identified by vacuolar degeneration in four cases; areas of undulated and/or fragmented myocardial fibers, with eosinophilia and nuclear pyknosis with or without enucleation of the myocytes in nine cases; and in one case, we observed a large area of myocardial necrosis. Immunohistochemical criteria confirmed the presence of the SARS-CoV-2 antigen at the level of the myocardium in only two cases. Comorbidities existing prior to SARS-CoV-2 infection associated with systemic and local inflammatory, thrombotic, hypoxic, or immunological phenomena influence the development of cardiac lesions, leading to death.

## 1. Introduction

Cardiovascular injury in the context of SARS-CoV-2 infection plays an essential role in the evolution of the disease, secondary only to pulmonary injury. Patients with pre-existing cardiovascular disease and SARS-CoV-2 infection have a high mortality rate (10.5%) [1,2]. Cardiovascular comorbidities are responsible for two-thirds of intensive-care unit (ICU) admissions, followed by diabetes, obesity, chronic kidney disease, age over 65 years, and cancer among patients hospitalized with SARS-CoV-2 infection.

Myocardial injury in patients with SARS-CoV-2 infection may be attributed to the presence of the virus at the cellular level. However, myocardial injury may also be secondary to pulmonary thromboembolism, arrhythmias, coronary thrombotic obstruction, congestive heart failure, sepsis, or disseminated intravascular coagulation. Using transthoracic echocardiography, other researchers have identified, as a primary manifestation, the presence of myocardial dysfunction in 70% of patients hospitalized with SARS-CoV-2 infection, similar to pulmonary damage [3]. The binding of SARS-CoV-2 to ACE2 (angiotensin-converting enzyme 2) receptors in pericytes is associated with microvascular dysfunction and dislocation of atherosclerotic plaque responsible for acute coronary syndrome [4]. In patients with a history of heart failure, increased expression of ACE2 receptors explains the risk of enhanced viral infectivity and increased mortality [5]. SARS-CoV-2 causes a cytokine storm that results in the release of large amounts of pro-inflammatory cytokines (IL-6, IL-7, IL-22, and CXCL10) and the activation of T-lymphocytes and macrophages that infiltrate the infected myocardium, which is associated with fulminant myocarditis and severe heart failure. The entry of SARS-CoV-2 into myocytes is directly responsible for severe myocardial dysfunction and arrhythmias such as atrial fibrillation, sinus tachycardia, and atrial or ventricular flutter. The appearance of cardiac arrhythmias in patients with SARS-CoV-2 infection, considered the second most severe event after acute respiratory distress syndrome (ARDS), is the result of metabolic disorders, cytokine storm, myocardial inflammation, and activation of the sympathetic nervous system [6]. Arrhythmias are present in 44% of SARS-CoV-2 patients admitted to the ICU and 7% of patients admitted to other units [5]. Even in the presence of mild clinical manifestations, such as palpitations or chest discomfort, electrocardiogram (ECG) recordings confirms the presence and severity of myocarditis [7].

In hospitalized patients, SARS-CoV-2 infection is associated with various degrees of cardiac injury—approximately 20% of cases demonstrate changes in high-sensitivity troponins [4] and 36% of cases demonstrate increases in troponin I within 24 hours of admission [8]. When investigated, cardiac injury is confirmed either by increases in troponins levels, as previously described, or by cardiac magnetic resonance imaging [9,10,11]. There is little evidence in the literature that supports the direct cardiac injury caused by SARS-CoV-2 [12,13].

In our clinical experience with hospitalized SARS-CoV-2 patients, we have most often encountered atrial fibrillation, sinus tachycardia, and atrial or ventricular flutter. Atrioventricular block and ventricular extrasystoles were observed less frequently. Similar to our Chinese colleagues [14], we evaluated patients presenting to our urgent care unit during the fifth SARS-CoV-2 wave with palpitations and chest pain, in the absence of previous respiratory manifestations. The diagnosis of SARS-CoV-2 infection was supported by a positive result of real-time reverse transcriptase-polymerase chain reaction (RT-PCR) assay from nasal and pharyngeal swabs, with respiratory manifestations developing after initial cardiac symptoms. Myocardial injury in the setting of SARS-CoV-2 infection can be attributed to several mechanisms including systemic and local inflammation, ischemia, hypoxia, hypertension, immunological mechanisms, direct viral action [15], and even antiviral medication toxicity. Immunological mechanisms trigger either a subclinical autoimmune myocarditis, secondary to a cytokine storm, or an autoimmune reaction as a consequence of myocardial injury.

## 2. Materials and Methods

Based on these observations regarding the mechanisms involved in cardiac injury in SARS-CoV-2 infection, we evaluated 16 patients who died as a result of SARS-CoV-2 infection, analyzing the group from both a clinical and pathological point of view. All patients were hospitalized in the County Clinical Emergency Hospital Sibiu, Romania, a county hospital with 1054 beds dedicated to the treatment of COVID-19 patients from the beginning of the pandemic. All autopsies were conducted in the Sibiu County morgue, taking into consideration all the national protocols for COVID-19 patients. Data collected included clinical aspects (past medical history, type of comorbidities) and postmortem macro and microscopic findings from a pathological viewpoint. The study was completed by immunohistochemical examination.

After tissue samples were obtained during autopsy, they were fixed in a 10% formalin solution. After fixation and dehydration using formalin, toluene, and different concentrations of methilic alcohol, the samples were embedded in paraffin blocks. Using a microtome, paraffin-embedded tissue was cut and mounted on microscopic slides. After deparaffinization, the slides were stained with the classic hematoxylin-eosin colorant.

For the immunohistochemical analysis, the paraffin-embedded tissue was cut at 3–5 microns and mounted on microscopic positively charged slides. The slides were air-dried for 2 h at 580 °C, then deparaffinized, dehydrated, and rehydrated. The tissues were then subjected to heat-induced epitope retrieval (HIER) using ImmunoDNA Retriever (Bio SB, Santa Barbara, CA, USA) with citrate. After heat treatment, the slides were transferred to an ImmunoDNA Retriever (Bio SB, Santa Barbara, CA, USA) with citrate buffer at room temperature for 15 minutes. For the antibody incubation, we used an automatic IHC method with the help of Epredia Autostainer 360 (epredia, Kalamazoo, MI, USA) using a mouse monoclonal antibody, IgG2b isotype, to detect the nucleocapsid SARS-CoV-2 antigen within the cytoplasm of the infected paraffin-embedded tissue cells. For the positive control, we used lung tissue infected in vitro. We obtained two negative controls, one using myocardial tissue from the same patients but without antibody incubation, and another using tissue from a confirmed negative PCR heart tissue using the same protocol as above.

Photographs were taken with the help of a Leica ICC50 W (Leica Microsystems GmbH, Wetzlar, Germany) microscope camera mounted on a Leica DM500 (Leica Microsystems GmbH, Wetzlar, Germany) microscope.

## 3. Results

Demographic and main clinical characteristics of the 16 enrolled patients are shown in Table 1. Two female and fourteen male patients with a mean age of 64.18 years (ranging from 35 to 80 years, standard deviation 22.5 years) were enrolled in the study.

Of the 16 patients included in this study, six patients presented with hypertension; five had congestive heart failure; one had chronic atrial fibrillation; two had a history of coronary artery bypass surgery; eight patients presented with type 2 diabetes mellitus, four of whom had hypertension; four had chronic kidney disease; one developed acute kidney failure; three had a neoplasm diagnosis; two had chronic lymphocytic leukemia; one had multiple sclerosis; and seven were obese. Fourteen patients were clinically and radiologically confirmed to have SARS-CoV-2 pneumonia.

Of the 16 autopsies we performed, two were complete, including an extensive examination of the cranial cavity. In 12 autopsies, we examined only the thoracoabdominal and pelvic cavities, and the remaining two autopsies were minimally invasive for the sole purpose of collecting tissue samples for histopathological and immunohistochemical examinations. All autopsies were performed in the COVID-19 morgue restricted area of the hospital, using complete protective equipment according to the current legal stipulations.

The primary cause of death (Table 2) for nine patients was severe acute respiratory distress following extensive lung damage secondary to viral pneumonia, in some cases with bacterial superinfection. In three cases, we found complete blockage of the pulmonary arterial trunk or blockage of one pulmonary artery, resulting in a massive pulmonary embolism; two of these patients were admitted less than 14 days prior to death. Two patients died of peritonitis—one secondary to perforated diverticulitis and the other secondary to a previously unknown perforated cecal adenocarcinoma. Two deaths did not occur during the acute phase of SARS-CoV-2 infection. The primary cause of death for these patients was chronic respiratory insufficiency due to diffuse pulmonary fibrosis following SARS-CoV-2 viral pneumonia.

Liver changes were detected in 14 patients in the form of hepatic steatosis and fibrosis. Acute renal injury was observed in four cases, chronic kidney damage (such as nephroangiosclerosis) in eight cases, and congenital horseshoe kidney in one case. In eight cases, we detected a degree of generalized arterial atherosclerosis in vessels including the aorta and coronary arteries. In the two cases in which the cranial cavity was opened, the only macroscopic finding was varying degrees of cerebral edema.

Macroscopic cardiac examination of our 16 postmortem patients revealed no morphological abnormalities in the pericardium or pericardial cavity. The most consistent findings were chronic in nature, such as left ventricular hypertrophy and varying degrees of coronary artery disease, with two cases having evidence of prior coronary angioplasty. In seven cases, dilation of the right heart chambers was observed, suggesting a form of pulmonary hypertension. Examination of the three cases demonstrated cardiac flaccidity, which may indicate the degree of acute cardiac insufficiency. The most significant findings were present in two cases that demonstrated lesions upon the cut section of the myocardium in the form of diffuse hyperemic territories alternating with pale areas, punctiform microhemorrhages, and a well-circumscribed hemorrhagic patch with a faded halo around its border. These findings may be indicative of severe circulatory injury, myocarditis, or diffuse myocardial damage of viral origin. Macroscopic aspect of the heart is presented in Figure 1.

### 3.1. Microscopic Cardiac Examination of Our 16 Postmortem Patients Revealed the following Changes



**Chronic alterations of the heart**
-Fatty infiltration of the myocardium: five cases-Focal or diffuse areas of fibrosis interposed between myocardial fibers, sequestrating groups, or individual myocytes: 11 cases. In two cases, we observed the appearance of granulation tissue, defined by newly formed blood vessels along with some scattered lymphocytes-Severe atherosclerosis of the coronary arteries: two cases

**Acute inflammatory and vascular lesions**
-Lymphocytic inflammatory infiltrate in the subepicardial region and/or visceral pericardium: nine cases-Marked vascular congestion with large numbers of neutrophils within the vascular lumen: five cases-Vascular thrombosis: three cases-Marked vascular congestion: three cases

**Acute ischemic or apoptotic injury (myocytic suffering)**
-Vacuolar degeneration of myocytes: four cases-Areas of undulated and/or fragmented myocardial fibers, with eosinophilia and nuclear pyknosis with or without enucleation of the myocytes: nine cases-In one case (with positivity for SARS-CoV-2 antibody), we observed a large area of myocardial necrosis with enucleated muscle fibers, eosinophilia, extravasated erythrocytes, scattered lymphocytes, neutrophils, and necrotic debris, with a halo of granulation tissue composed of newly formed capillaries, macrophages, and lymphocytic infiltrate.



Microscopic aspects of the heart are presented in Figure 2, Figure 3 and Figure 4.

Detailed information regarding the macroscopic and microscopic cardiac examination is reported in Table 3 and Table 4, respectively.

### 3.2. Immunohistochemical Study

The study includes all 16 cases, investigating cardiac tissue collected from all significant areas of myocardium within the autopsies.

Two of the 16 cases presented focal positivity for the SARS-CoV-2 antibody. In one case (case no. 11) the positive cells were macrophages and fibroblasts within a patch of myocardial injury with myocytic fragmentation, eosinophilia, nuclear pyknosis, or enucleosis with scattered lymphocytic infiltrate, but with no positivity for the remaining myocardial fibers in this area. In another case (case no. 6), there was a large area of myocardium that weakly stained positive, and between these, patches of myocardial fibers there were small sectors of myocardial injury evidenced by myocytic necrosis with enucleated muscle fibers, eosinophilia, extravasated erythrocytes, scattered lymphocytes and neutrophils, and necrotic debris, as well as granulation tissue composed of newly formed capillaries, macrophages, and lymphocytic infiltrate. In these sectors, we observed intense positivity in the cytoplasm of the macrophages and fibroblasts. At the periphery of these areas, there were histologically normal myocardial fibers that presented focal cytoplasmic positivity for our antibody (Figure 5, Figure 6 and Figure 7).

## 4. Discussion

As of 2020, SARS-CoV-2 has been added to the list of viral etiologies associated with cardiac injury, along with Coxsackie viruses, adenoviruses, parvovirus B19 [16], influenza virus, and HIV. According to the National Health Commission of the People’s Republic of China [17], myocardial injury in the setting of SARS-CoV-2 infection is associated with the presence of myocyte necrosis and inflammatory infiltrates.

The presence of ACE2 co-receptors [18] and of TMPRSS2 (transmembrane serine protease 2) [19] both in the lungs and in the gastrointestinal, renal, cardiovascular, and nervous systems, explains the damage of these systems in SARS-CoV-2 infection as well as the interconnection between the changes of these systems [20].

Changes in the gut microbiota can increase the lung immune response and vice versa, and changes in the lungs cause alterations in the flora of the digestive tract. SARS-CoV-2 virus, in addition to the inflammatory phenomena of the colon and the changes in the gut microbiota, is also responsible for α-synuclein upregulation. An increase in the level of proinflammatory cytokine causes increases in colonic permeability, secondary increases in serum lipopolysaccharide levels, and the synthesis of proinflammatory cytokines in the central nervous system [21]. If we consider the disruption of the integrity of the digestive barrier, present during comorbidities associated with severe or critical forms of COVID-19 (diabetes, obesity, and hypertension, as well as other cardiovascular diseases and malignancies), the facilitated access of SARS-CoV-2 virus at ACE2 receptors, as well as changes in the ACE/ACE2 ratio, is understandable. Changes in the gut microbiota increase the release of Th17 (T helper 17), which accentuates the lung lesions and favors the appearance of autoimmune phenomena on the gut long axis [22].

The gut–brain axis may also explain the presence of digestive symptoms such as nausea and vomiting, associated with headache, as well as the hypertension syndrome, suggesting the presence of the virus in the central nervous system (sometimes confusing symptoms as belonging to one or the other of the two systems). Inflammation of the digestive tract can trigger cognitive changes via the vagus nerve. Histologically, the presence of ACE2 and TMPRSS2 receptors was found abundantly expressed in enteric neurons, intestinal glial cells, the meningocerebral barrier, and the choroid plexuses [23].

Hepatic impairment in SARS-CoV-2 infection may be associated with increased ACE2 receptor expression, the presence of other co-receptors, hypoxia with secondary ischemia, sepsis, drug-induced hepatotoxicity, or the presence of chronic liver disease, such as chronic hepatitis B infection. Severe evolution of COVID-19 cases is associated with both heart damage, septic shock, and acute or severe hepatic impairment aggravated in an infectious setting [24].

Returning to the anatomopathological aspects, Linder et al. [12] performed 39 autopsies on patients who died of SARS-CoV-2 infection and stated that there were no differences between inflammatory infiltrates in patients with documented cardiac infection versus those without cardiac infection. The authors stated that the SARS-CoV-2 virus is not found in cardiomyocytes but rather in interstitial cells or macrophages present in the myocardium. In this German study, the presence of the SARS-CoV-2 virus was proven in 61.5% of the samples, 41% of which had a high viral load (>1000 copies/μg); the presence of the viral replication was detected in five of the 16 patients with a high viral load.

Examining 16 cases of deaths due to SARS-CoV-2 infection, Kawakami et al. identified the presence of the SARS-CoV-2 virus in the myocardium in only two cases [25]. Our study also showed the presence of the virus in the myocardium in two out of 16 cases. It is possible that the leukocytopenia from SARS-CoV-2 infection may explain the absence of mononuclear cells in the myocardium [26].

Pathologically, the most consistent finding in our studied cases of acute heart injury was a prominent lymphocytic inflammatory infiltrate in the subepicardial space and visceral pericardium, which was found in 9 out of 16 cases, indicating a form of microscopic lymphocytic pericarditis. Two of these patients were between 30 and 39 years of age, and apart from one of them being obese, they presented no other comorbidities. Two patients had congestive heart failure, one of whom had severe coronary atherosclerotic blockage with a history of coronary artery by-pass surgery. Although we cannot demonstrate the exact pathogenic mechanism of this discovery, we feel it is necessary to report these types of findings so overall we can elucidate how the virus interacts with different tissues and, of course, explain the long-covid syndrome observed in some patients. Another important finding was acute ischemic injury present in 9 out of 16 patients, translated by areas of undulated and/or fragmented myocardial fibers, with eosinophilia, nuclear pyknosis, and anucleation of the myocytes with or without focal myocardial necrosis unrelated to age, gender, or comorbidities. In at least two cases, we concluded, after immunohistochemical analysis, that the ischemic injury was caused by direct viral infection of the myocytes, macrophages, and fibroblasts within the myocardial tissue. In the other six cases, acute ischemic injury could have been initiated by the hemodynamic instability secondary to acute respiratory distress syndrome. Small blood vessel thrombosis of the myocardium was found in three cases, a consistently described injury in the lung parenchyma. An interesting aspect that we found in 11 cases was focal or diffuse areas of fibrosis, another consistently described injury of the lungs of patients who had recently suffered from a moderate to severe form of SARS-CoV-2 pneumonia. The exact mechanism by which, in some patients, the fibrotic process is activated after infection has not been described in the medical literature to date. Unfortunately, we cannot demonstrate in this study the exact time that the myocardial fibrotic process started and whether it was connected to the COVID-19 infection. Interestingly, these patches of fibrosis were also found in three cases of patients under 45 years of age without any known cardiac comorbidities, suggesting the possibility of a recent fibrotic process activation, possibly associated with the SARS-CoV-2 infection. In other cases, fibrosis was associated with chronic ischemic cardiac disease; therefore, the association between this alteration and the SARS-CoV-2 infection could not be demonstrated.

One positive cardiac tissue for SARS-CoV-2 antibody (case no. 6) came from a 68-year-old man with a history of hypertension, diabetes, and obesity (Table 1) who was admitted 1 day prior to his death. This is the same case that presented with the most significant macroscopic and microscopic lesions described earlier (Table 3 and Table 4). The other case that was positive for our antibody (case no. 11) came from a 58-year-old man with a history of diabetes and chronic lymphocytic leukemia (Table 1), and chronic ischemic cardiomyopathy was diagnosed after the autopsy was performed (Table 3). Histopathology revealed the presence of acute myocardial injury in this case (Table 4). The patient was admitted 10 days prior to his death. Both patients were diagnosed clinically and radiologically with severe viral pneumonia.

Our study had some limitations. The main limitation of our study was the small population of enrolled patients in the study and the small number of performed autopsies. This was a single-center study.

## 5. Conclusions

In conclusion, according to our results, cardiac injury in the context of SARS-CoV-2 infection is multifactorial. The direct action of SARS-CoV-2 was present in only 12.5% of the cases. Prior comorbidities associated with systemic and local inflammatory, thrombotic, hypoxic, or immunological phenomena influenced the development of cardiac lesions during SARS-CoV-2 infection leading to death.

## Figures and Tables

**Figure 1 microorganisms-10-01258-f001:**
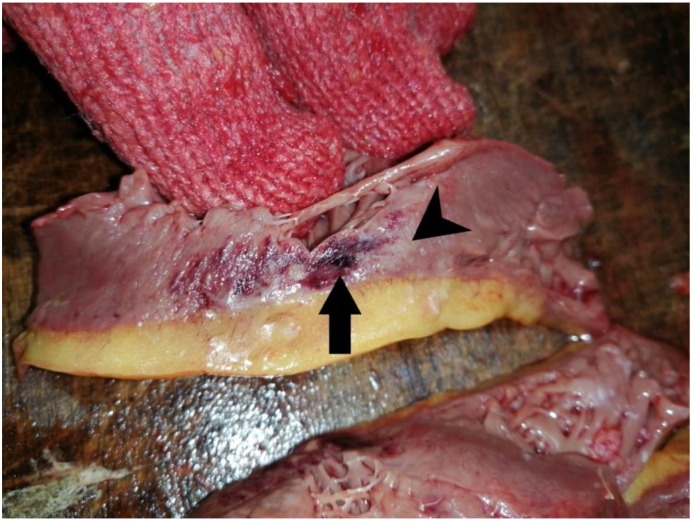
Macroscopic aspect of heart: well circumscribed hemorrhagic patch (arrow) with a pale halo (arrowhead).

**Figure 2 microorganisms-10-01258-f002:**
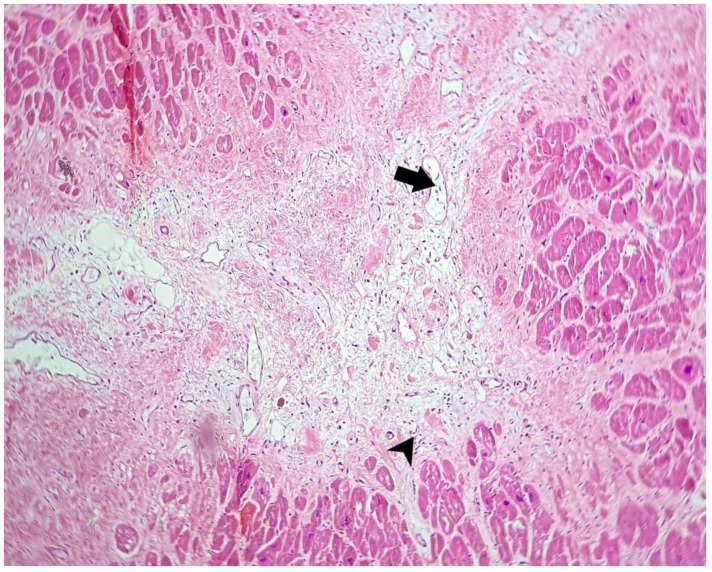
Microscopic aspect of heart using a hematoxylin-eosin stain: fibrosis with a myxoid appearance, with an increased number of blood vessels (arrow) and a perivascular scattered lymphocytic infiltrate (arrowhead) (100×).

**Figure 3 microorganisms-10-01258-f003:**
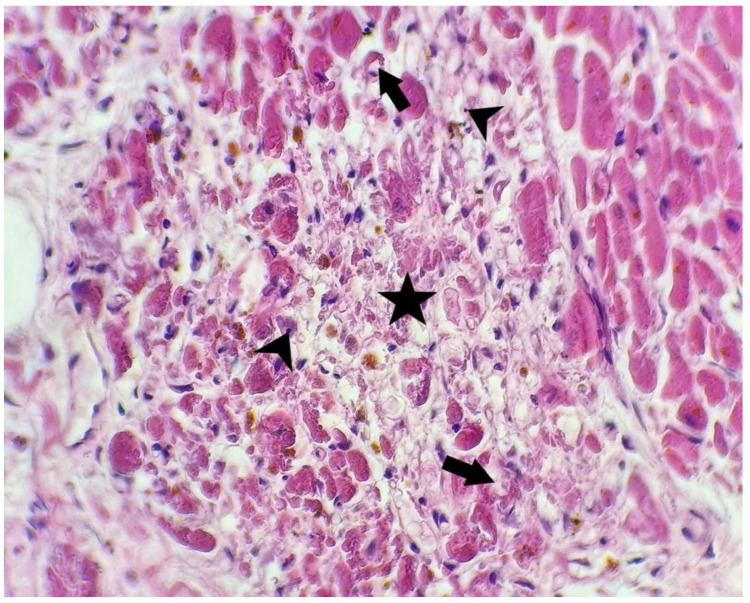
Microscopic aspect of heart using a hematoxylin-eosin stain: myocardic fibers with cytoplasmic vacuolation (arrows), nuclear piknosis (arrowheads), and necrosis (star) (400×).

**Figure 4 microorganisms-10-01258-f004:**
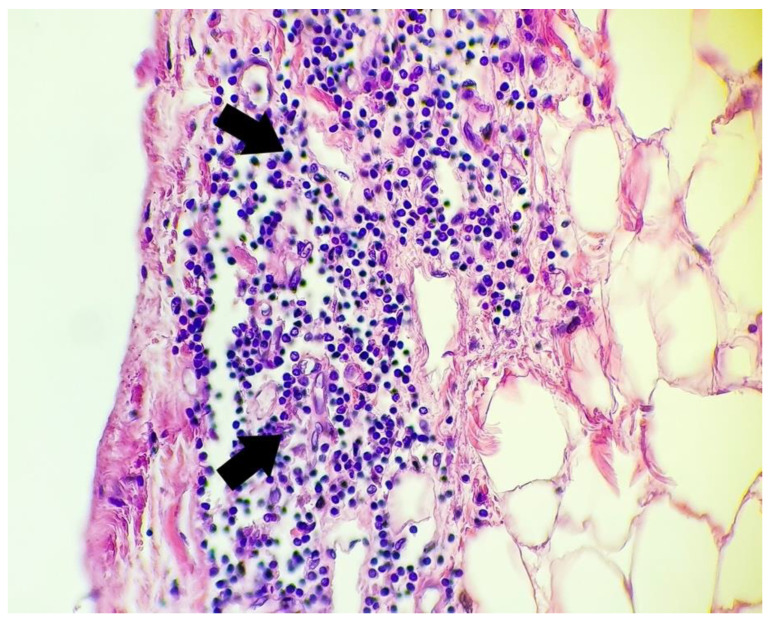
Microscopic aspect of heart using a hematoxylin-eosin stain: rich diffuse inflammatory lymphocytic infiltrate (arrows) in the subepicardial space (400×).

**Figure 5 microorganisms-10-01258-f005:**
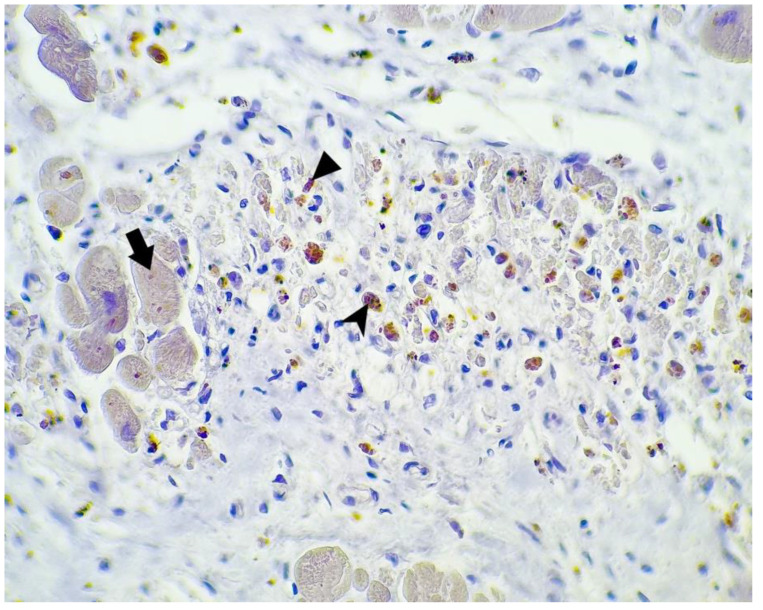
SARS-CoV-2 antibody immunohistochemistry; myocardium: weakly positive staining for SARS-CoV-2 of the myocardic fibers (arrow) and intense positivity in the cytoplasm of the macrophages (arrowhead) and fibroblasts (triangle) (400×).

**Figure 6 microorganisms-10-01258-f006:**
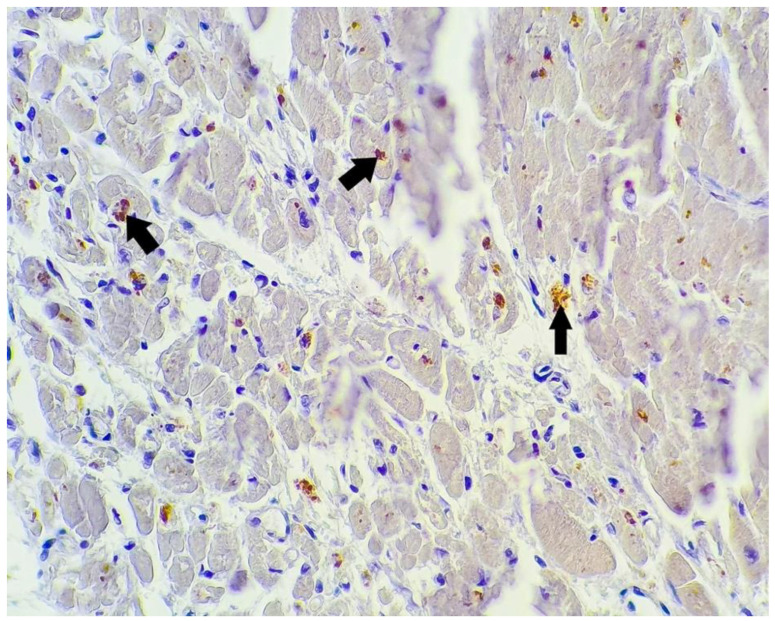
SARS-CoV-2 antibody immunohistochemistry; myocardium: myocardic fibers with focal cytoplasmic positivity for SARS-CoV-2 antibody (arrows) (400×).

**Figure 7 microorganisms-10-01258-f007:**
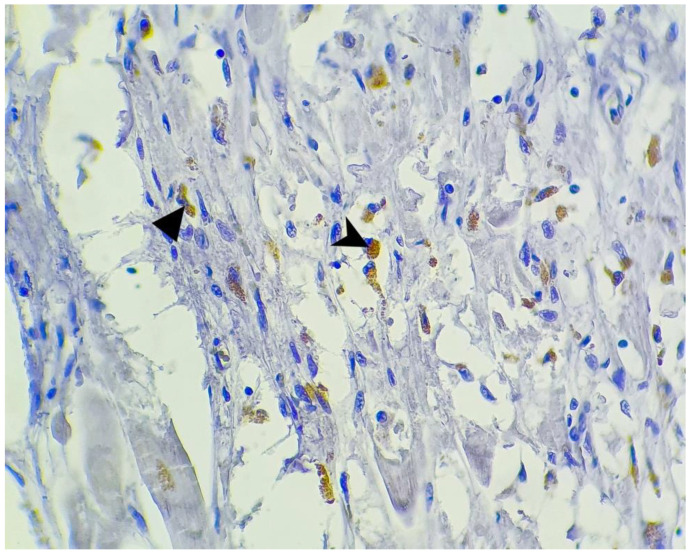
SARS-CoV-2 antibody immunohistochemistry; myocardium: positive macrophages (arrowhead) and fibroblasts (triangle) within a patch of myocardic injury (400×).

**Table 1 microorganisms-10-01258-t001:** Demographic data and comorbidities.

Gender	Age (Years)	Hypertension	Heart Failure	Chronic Atrial Fibrillation	Coronary Artery Bypass Surgery	Diabetes	Chronic Kidney Disease	Obesity	Pneumonia	Hematological Disease/Neoplasm
M	35								Y	
M	79									
M	72	Y	Y			Y	Y		Y	Anemia
M	43								Y	
M	50					Y		Y	Y	
M	68	Y				Y		Y	Y	
M	61								Y	
M	80								Y	Liver and bone metastases with unknown point of origin
M	35							Y	Y	
M	79	Y	Y	Y	Y	Y	Y	Y	Y	
M	58					Y				Chronic lymphocytic leukemia
M	74	Y				Y		Y	Y	Multiple sclerosis
M	68		Y					Y	Y	
M	77	Y	Y		Y	Y	Y		Y	Rectal, bladder tumor
M	72	Y	Y		Y		Acute injury		Y	Lymphocytic leukemia
F	76					Y	Y	Y	Y	Cecum tumor

Y-yes.

**Table 2 microorganisms-10-01258-t002:** Primary cause of death.

Cause of Death	<14 Days Since Admission	>14 Days Since Admission	Total
Acute respiratory distress secondary to pneumonia	8 cases	1 case	9 cases
Pulmonary thromboembolism	2 cases	1 case	3 cases
Peritonitis	1 case	1 case	2 cases
Chronic pulmonary insufficiency secondary to diffuse fibrosis	-	2 cases	2 cases

**Table 3 microorganisms-10-01258-t003:** Macroscopic cardiac examination, details.

CASE	Macroscopic Cardiac Examination
Case no. 1	-
Case no. 2	Right and left ventricular hypertrophy Myocardosclerosis Atherosclerosis of the coronary arteries
Case no. 3	-
Case no. 4	Flaccid cardiac consistency Right atrial and ventricular dilatation Diffuse hyperemic areas alternating with pale areas on the myocardial cut surface Punctiform microhemorrhages on cut surface
Case no. 5	Right atrial and ventricular dilatation Mild left ventricular hypertrophy
Case no. 6	Right atrial and ventricular dilatation Severe left ventricular hypertrophy Diffuse hyperemic areas alternating with pale areas on the myocardial cut surface Well-circumscribed hemorrhagic patch with a pale halo on the cut surface of the inferior septal wall of the myocardium
Case no. 7	Flaccid cardiac consistency Right atrial and ventricular dilatation Left ventricular hypertrophy Atherosclerosis of the coronary arteries
Case no. 8	Right atrial and ventricular dilatation Left ventricular hypertrophy Myocardosclerosis Atherosclerosis of the coronary arteries
Case no. 9	Diffuse hyperemic areas
Case no. 10	Dilated cardiomyopathy Fibrous scar at the anterior-inferior margin of the left ventricle Myocardosclerosis Atherosclerosis of the coronary arteries
Case no. 11	Myocardosclerosis Atherosclerosis of the coronary arteries
Case no. 12	Left ventricular hypertrophy Anterior descendent coronary artery angioplasty
Case no. 13	Right atrial and ventricular dilatation Left ventricular hypertrophy
Case no. 14	Dilated cardiomyopathy Flaccid cardiac consistency Atherosclerosis of the coronary arteries
Case no. 15	Dilated cardiomyopathy Atherosclerosis of the coronary arteries Multiple coronary angioplasties
Case no. 16	Right atrial and ventricular dilatation Left ventricular hypertrophy

**Table 4 microorganisms-10-01258-t004:** Microscopic Cardiac Examination, details.

CASE	Microscopic Cardiac Examination
Case no. 1	Epicardial fat with fatty infiltration of the myocardium Small patches of fibrosis in the subendocardial region with surrounding scattered inflammatory cells Vacuolar degeneration of the myocytes in the subendocardial regions Small areas of fragmented myocardial fibers, with eosinophilia and nuclear pyknosis Scattered lymphocytic inflammatory infiltrate in the subepicardial region
Case no. 2	Large extended areas of fibrosis interposed between myocardial fibers, sequestrating groups, or individual myocytes. Patchy inflammatory lymphocytic infiltrate in subepicardial space and visceral pericardium. Small areas of fragmented myocardial fibers, with eosinophilia and nuclear pyknosis; in some parts enucleation of the myocytes Small areas of scattered neutrophils and extravasated erythrocytes replacing the myocardial fibers
Case no. 3	Focal areas of fibrosis interposed between myocardial fibers, sequestrating groups, or individual myocytes Vacuolar degeneration of scattered myocytes Small number of neutrophils in the perivascular region of some medium size arterial vessels Scattered lymphocytic inflammatory infiltrate in the subepicardial region
Case no. 4	Small areas of fatty infiltration of the myocardium. Vascular congestion with numerous intraluminal neutrophils, some of which extravasate in the interstitium along with some lymphocytes Marked vascular congestion Limited area of prominent lymphocytic infiltration of the vascular wall of the large and medium size vessels with edema and endothelial erosions Small areas of fragmented and/or undulated myocardial fibers with eosinophilia and nuclear pyknosis; in some parts enucleation of the myocytes Patchy inflammatory lymphocytic infiltrate in the subepicardial space and visceral pericardium Small patches of fine fibrotic bands interposed between myocardial fibers. In these areas there is also fatty infiltration and capillarization along with some scattered lymphocytes (suggesting a recent acute injury in the reparative phase)
Case no. 5	Microthrombi formation in the small blood vessels within the myocardium Small areas of undulation of the myocardial fibers
Case no. 6	Rich diffuse inflammatory lymphocytic infiltrate in the subepicardial space visceral pericardium, with focal fibrin deposits on the surface Large extended areas of fibrosis interposed between myocardial fibers, sequestrating groups or individual myocytes, with thick, poorly cellularized, horizontally-arranged collagen fibers. In these patches of fibrosis, there is a more myxoid appearance, with an increased number of blood vessels and with a perivascular scattered lymphocytic infiltrate Small, poorly-defined areas of myocardial fibers with cytoplasmic vacuolation, nuclear pyknosis, apoptosis, and necrotic debris between these fibers Undamaged myocytes near the fibrotic patches suffered nuclear enlargement, with irregular borders and granular fine dispersed chromatin, a sign of cellular suffering Severe atherosclerosis of the coronary arteries with secondary subocclusion Large area of myocardial necrosis with enucleated muscle fibers, eosinophilia, extravasated erythrocytes, scattered lymphocytes and neutrophils and necrotic debris, with a halo of granulation tissue composed of newly-formed capillaries, macrophages, and lymphocytic infiltrate
Case no. 7	Small and medium-sized vessel thrombosis Small areas of scattered lymphocytic and neutrophilic inflammatory infiltrate in the subepicardial region
Case no. 8	Small patches of fibrosis, more predominantly in the perivascular region Small areas of fragmented and/or undulated myocardial fibers with eosinophilia. Scattered lymphocytic inflammatory infiltrate in the subepicardial region
Case no. 9	Small patches of fibrosis in the subpericardial region Small patches of fibrosis in the subendocardial region Small areas of fatty infiltration of the myocardium Undulation of the myocardial fibers Myocardial fiber fragmentation with nuclear pyknosis and enucleation Marked vascular congestion with large numbers of neutrophils within the vascular lumen Patchy inflammatory lymphocytic infiltrate in the subepicardial space and visceral pericardium. Small number of extravasated lymphocytes in the perivascular space Subendocardial necrosis with eosinophilia, enucleation, vacuolization, and cytoplasmic intumescence of the myocytes
Case no. 10	Extended diffuse fibrosis interposed between large groups of myocardial fibers, in a transmural fashion Myocardial fiber hypertrophy Marked vascular congestion Severe atherosclerosis of the coronary arteries with partial obstruction and calcification
Case no. 11	Patch of myocardial injury with myocytic fragmentation, eosinophilia, nuclear pyknosis, or enucleation with scattered lymphocytic infiltrate Vacuolation and eosinophilia of myocardial fibers near the subepicardial space.
Case no. 12	Vascular congestion Focal areas of fibrosis interposed between myocardial fibers, sequestrating groups, or individual myocytes
Case no. 13	Marked vascular congestion with large numbers of neutrophils within the vascular lumen Small patches of fibrosis interposed between myocardial fibers, sequestrating groups, or individual myocytes, more prominent in the subendocardial region Fatty infiltration of the myocardium
Case no. 14	Small vessel thrombosis Fine fibrotic bands interposed between myocardial fibers with patches of fibrosis forming in the subendocardial region
Case no. 15	Rich diffuse lymphocytic infiltrate in the subepicardial region Fibrosis bands between groups of myocardial fibers Scattered small number of lymphocytes in the perivascular region
Case no. 16	Fatty infiltration of the myocardium. Eosinophilia, myocardial fragmentation, and undulation

## Data Availability

All data generated or analyzed during this study are included in this published article.

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
