# Peer review of "Is SARS-CoV-2 Directly Responsible for Cardiac Injury? Clinical Aspects and Postmortem Histopathologic and Immunohistochemical Analysis"

_microorganisms, 2022, doi:10.3390/microorganisms10071258_

Round 1

Reviewer 1 Report

The manuscript “Is SARS-CoV-2 directly responsible for cardiac injury? Clinical aspects and 3 postmortem histopathologic and immunohistochemical analysis" present the morphological cardiac tissues changes in patients with COVID-19.

As it is structured, the manuscript cannot be accepted for publication.

Comment 1: In the Introduction the sentences must have some connection with each other.

Comment 2: Materials and Methods must be detailed, regarding processing for histology and immunohistochemistry (IHC) and how the photographs were taken. The subchapter presented by the authors regarding IHC (lines 210-212) should be part of the M&M and should include the protocol, whether HIER was performed, negative controls, etc.

Comment 3: In the Results, the reference to tables and figures must be indicated in the text. For each case, the gender and age must be indicated in order to have an idea of the changes that may be associated with these analysis criteria. Following the authors' logic (for example macroscopic changes), the information concerning Histopathology (lines 186-208) and IHC (lines 215-226) could also be presented in table form. In the impossibility of having access to slides of heart tissue from patients with some of the comorbidities, which would somehow work as a control, the authors should make this reference, or else, discuss a little this aspect in the Discussion section. The legends of Figures 1, 3 and 4 should also include notations (for example, arrows) of the findings.

Comment 4: Authors should deepen the Discussion, including some information concerning heart tissue changes from patients with some of the comorbidities. The final part of the Discussion turns out to be a repetition of the description of the Results.

Comment 5: Although the study was approved by the Ethics, Medical Ethics and Deontology Committee of the County Clinical Emergency Hospital Sibiu (No. 22893/2020/IX/22), no reference is made to informed consent from patients' family

Author Response

Sibiu, 13.06.2022

To

the Editors of Microorganisms ®

Dear Editor,

Dear reviewer,

Thank you for reviewing our manuscript. Please find attached a revised version of our manuscript, “Is SARS-CoV-2 directly responsible for cardiac injury? Clinical aspects and postmortem histopathologic and immunohistochemical analysis”.

Yours and the reviewers’ comments were highly insightful and enabled us to greatly improve the quality of our manuscript. We have modified the manuscript in response to the comments. Attached is our point-by-point response to each comment.

Reviewer Comments:

The manuscript “Is SARS-CoV-2 directly responsible for cardiac injury? Clinical aspects and 3 postmortem histopathologic and immunohistochemical analysis" present the morphological cardiac tissues changes in patients with COVID-19. 

As it is structured, the manuscript cannot be accepted for publication. 

A: The manuscript has been prepared according to the Microsoft Word template. As it is mentioned in the instructions for authors sections “Accepted File Formats - Authors are encouraged to use the Microsoft Word template or LaTeX template to prepare their manuscript.” As it is not mandatory we prefer that the copy-editing of the manuscript be performed by the editorial office after the manuscript is accepted.

Comment 1: In the Introduction the sentences must have some connection with each other.

A: The paragraphs of the section of the manuscript have been rearranged to fulfill the issue.

Comment 2: Materials and Methods must be detailed, regarding processing for histology and immunohistochemistry (IHC) and how the photographs were taken. The subchapter presented by the authors regarding IHC (lines 210-212) should be part of the M&M and should include the protocol, whether HIER was performed, negative controls, etc.

A: A full description of the histopathological examination and the immunohistochemical analysis was included in the material and method section of the manuscript.

Comment 3: In the Results, the reference to tables and figures must be indicated in the text. For each case, the gender and age must be indicated in order to have an idea of the changes that may be associated with these analysis criteria. Following the authors' logic (for example macroscopic changes), the information concerning Histopathology (lines 186-208) and IHC (lines 215-226) could also be presented in table form. In the impossibility of having access to slides of heart tissue from patients with some of the comorbidities, which would somehow work as a control, the authors should make this reference, or else, discuss a little this aspect in the Discussion section. The legends of Figures 1, 3 and 4 should also include notations (for example, arrows) of the findings.

A: Demographic data are reported in table 1. The legend and the mentioned figures were updated. The tables are mentioned in the text. The figures are inserted in the text in their correct positions. We don’t consider that the HP and IHC studies to be reported in a table. The objective of this study was to report the histopathological and IHC aspects of the heart in COVID-19 patients and not the aspect from non-COVID-19 patients with comorbidities, these aspects are widely published in the literature. Our anatomopathologists can describe the microscopic aspects of the tissue samples.

Comment 4: Authors should deepen the Discussion, including some information concerning heart tissue changes from patients with some of the comorbidities. The final part of the Discussion turns out to be a repetition of the description of the Results.

A: The final part of the discussion section of the manuscript has been rewritten.

Comment 5: Although the study was approved by the Ethics, Medical Ethics and Deontology Committee of the County Clinical Emergency Hospital Sibiu (No. 22893/2020/IX/22), no reference is made to informed consent from patients' family

A: The “Informed Consent Statement” section of the manuscript was updated with the following information: The autopsy agreement signed by the patients’ families contains a paragraph that stipulates that the undersigned approves the use of medical information, photographic images of tissues and organs, including microscopy, under anonymity for scientific purposes.

(Not included in the manuscript - Government decision nr. 451/2004 for the approval of the methodological norms of Law number 104/2003 on the handling of human corpses and the removal of organs and tissues from corpses for transplantation stipulates in Article 6 paragraph c) that “the patient, his relatives or the legal representative, as the case may be, requests the expression of agreement, in writing for medical studies; didactic/scientific use of photographic images of tissues/organs harvested/examined.)

We hope that the revised form of the manuscript and our accompanying responses will be sufficient to make our manuscript suitable and accepted for publication. We shall look forward to hearing from you at your earliest convenience.

With our best regards,

Sincerely yours,

Rares Mircea Birlutiu, MD PhD

Victoria Birlutiu, Assoc. Prof. MD. PhD

Reviewer 2 Report

The topic of this manuscript falls within the scope of Microorganisms Journal.  In my opinion the topic of the presented paper is original and very important

There are some comments in the reviewer's opinion which should be taken under consideration by the Authors:

  1. Please use the Microsoft Word template or LaTeX template to prepare your manuscript;

https://www.mdpi.com/journal/microorganisms/instructions

2.      Please number the section

3.      Please read carefully the template and prepare the tables and their description and figures (Table1, etc)

4.       Immunohistochemical study section- the  method description please add to the section materials and method

5.      Please describe the section -materials and method in detail

6.      Please add the limitations of your study

7.      It has been mentioned that some of the patients had also lesions in the liver. Please discuss this issue. It has been postulated that the hepatic consequences of SARS-CoV-2 infection are an important problematic component of COVID-19 that is most important in patients with earlier liver disease who are at remarkably high risk of severe COVID-19 and death. SARS-CoV-2 enters the respiratory system and systemic circulation and accesses other systems, including the liver, GI, and CNS. The main mechanism of the entry of the virus into cells is mediated by the ACE-2 receptor, which is widely expressed in these systems. ( please cite:  a)Infection of Human Cells by SARS-CoV-2 and Molecular Overview of Gastrointestinal, Neurological, and Hepatic Problems in COVID-19 Patients. Journal of clinical medicine10(21), 4802. https://doi.org/10.3390/jcm10214802; b)Viral Infection-Induced Gut Dysbiosis, Neuroinflammation, and α-Synuclein Aggregation: Updates and Perspectives on COVID-19 and Neurodegenerative Disorders. ACS Chem Neurosci. 2020 Dec 16;11(24):4012-4016. doi: 10.1021/acschemneuro.0c00671;

c) SARS-CoV-2 pandemic and research gaps: Understanding SARS-CoV-2 interaction with the ACE2 receptor and implications for therapy. Theranostics. 2020 Jun 12;10(16):7448-7464. doi: 10.7150/thno.48076.

d) Angiotensin-converting enzyme 2 (ACE2), SARS-CoV-2 and the pathophysiology of coronavirus disease 2019 (COVID-19). J Pathol. 2020 Jul;251(3):228-248. doi: 10.1002/path.5471.

Author Response

Sibiu, 13.06.2022

To

the Editors of Microorganisms ®

Dear Editor,

Dear reviewer,

Thank you for reviewing our manuscript. Please find attached a revised version of our manuscript, “Is SARS-CoV-2 directly responsible for cardiac injury? Clinical aspects and postmortem histopathologic and immunohistochemical analysis”.

Yours and the reviewers’ comments were highly insightful and enabled us to greatly improve the quality of our manuscript. We have modified the manuscript in response to the comments. Attached are our point-by-point response to each comment.

Reviewer Comments:

The topic of this manuscript falls within the scope of Microorganisms Journal.  In my opinion the topic of the presented paper is original and very important

There are some comments in the reviewer's opinion which should be taken under consideration by the Authors:

Answer: Thank you for taking from your precious time to be able to assess our manuscript. The comments were highly insightful and enabled us to improve our manuscript.

Please use the Microsoft Word template or LaTeX template to prepare your manuscript;

https://www.mdpi.com/journal/microorganisms/instructions

A: Thank you for the suggestion. Done! The manuscript was prepared according to the Microsoft Word template.

  1. Please number the section

A: Thank you for the suggestion. Done!

  1. Please read carefully the template and prepare the tables and their description and figures (Table1, etc)

A: Thank you for the suggestion and for pointing this fact. We hope that we addressed all the issues.

  1. Immunohistochemical study section- themethod description please add to the section materials and method

A: A full description of the histopathological examination and of the immunohistochemical analysis was included in the material and method section of the manuscript. Thank you! The following statements were added: “After tissue samples were obtained during autopsy, they were fixed using 10% formalin solution. After fixation and dehydration using formalin, toluen and different concentrations of methilic alcohol, the samples were included in parrafin blocks. Using a microtome, the parrafin-embedded tissue was cut and mounted on microscopic slides. After deparrafination, the slides were stained with a classic hematoxilin-eosin colorant. For the immunohistochemical analysis, the parrafin-embedded tissue was cut at 3-5 microns and mounted on microscopic positively charged slides. The slides were air-dried for 2 hours at 580 C, then deparrafinazed, dehydrated, and rehydrated. The tissues were then subjected to heat induced epitope retrieval (HIER) using ImmunoDNA Retrievel with Citrate. After heat treatment, the slides were transferred into an ImmunoDNA Retrievel with citrate at room temperature for 15 minutes. For the antibody incubation, we used an automatic IHC method with the help of Epredia Autostainer 360 using a mouse monoclonal antibody, IgG2b isotype, for detection of the Nucleocapsid SARS-CoV-2 antigen within the cytoplasm of the infected paraffin embedded tissue cells. For positive control, we used in vitro infected lung tissue. We obtained two negative controls, one of them using myocardial tissue from the same patients but without antibody incubation and another using tissue from a confirmed negative PCR heart tissue using the same protocol as above. The photographs were taken with the help of a Leica ICC50 W Microscope Camera mounted on a Leica DM500 microscope.”

  1. Please describe the section -materials and method in detail

A: We hope that we addressed these issues with all the information that was added into the sections.

  1. Please add the limitations of your study

A: Done.

  1. It has been mentioned that some of the patients had also lesions in the liver. Please discuss this issue. It has been postulated that the hepatic consequences of SARS-CoV-2 infection are an important problematic component of COVID-19 that is most important in patients with earlier liver disease who are at remarkably high risk of severe COVID-19 and death. SARS-CoV-2 enters the respiratory system and systemic circulation and accesses other systems, including the liver, GI, and CNS. The main mechanism of the entry of the virus into cells is mediated by the ACE-2 receptor, which is widely expressed in these systems. ( please cite:a)Infection of Human Cells by SARS-CoV-2 and Molecular Overview of Gastrointestinal, Neurological, and Hepatic Problems in COVID-19 Patients. Journal of clinical medicine, 10(21), 4802. https://doi.org/10.3390/jcm10214802; b)Viral Infection-Induced Gut Dysbiosis, Neuroinflammation, and α-Synuclein Aggregation: Updates and Perspectives on COVID-19 and Neurodegenerative Disorders. ACS Chem Neurosci. 2020 Dec 16;11(24):4012-4016. doi: 10.1021/acschemneuro.0c00671;
  2. c) SARS-CoV-2 pandemic and research gaps: Understanding SARS-CoV-2 interaction with the ACE2 receptor and implications for therapy. Theranostics. 2020 Jun 12;10(16):7448-7464. doi: 10.7150/thno.48076.
  3. d) Angiotensin-converting enzyme 2 (ACE2), SARS-CoV-2 and the pathophysiology of coronavirus disease 2019 (COVID-19). J Pathol. 2020 Jul;251(3):228-248. doi: 10.1002/path.5471.

A: Thank you for the suggestions, we included the recommended references in the discussion section of the manuscript and discussed these issues.

We hope that the revised form of the manuscript and our accompanying responses will be sufficient to make our manuscript suitable and accepted for publication. We shall look forward to hearing from you at your earliest convenience.

With our best regards,

Sincerely yours,

Rares Mircea Birlutiu, MD PhD

Victoria Birlutiu, Assoc. Prof. MD. PhD

Round 2

Reviewer 1 Report

Authors should refer and report to readers, in the text of the Results section, Tables 3 and 4 and Figures 1, 2 3, 4, 5, 6 and 7. The reference to Tables 3 and 4 only appears in the Discussion and the authors never refer to the Figures, they simply appear in the end of the text.

Author Response

Sibiu, 15.06.2022

To

the Editors of Microorganisms ®

Dear Editor,

Dear reviewer,

Thank you for reviewing our manuscript. Please find attached a revised version of our manuscript, “Is SARS-CoV-2 directly responsible for cardiac injury? Clinical aspects and postmortem histopathologic and immunohistochemical analysis”.

Yours and the reviewers’ comments were highly insightful and enabled us to greatly improve the quality of our manuscript. We have modified the manuscript in response to the comments. Attached is our point-by-point response to each comment.

Reviewer Comments:

Authors should refer and report to readers, in the text of the Results section, Tables 3 and 4 and Figures 1, 2 3, 4, 5, 6 and 7. The reference to Tables 3 and 4 only appears in the Discussion and the authors never refer to the Figures, they simply appear in the end of the text.

Answer: We reassessed the manuscript for the reported issues and in the Results sections of the manuscript indicated in the text the references for tables and figures and also changed the positions of the figures.

We hope that the revised form of the manuscript and our accompanying responses will be sufficient to make our manuscript suitable and accepted for publication. We shall look forward to hearing from you at your earliest convenience.

With our best regards,

Sincerely yours,

Rares Mircea Birlutiu, MD PhD

Victoria Birlutiu, Assoc. Prof. MD. PhD